# Probabilistic Self-supervised Learning via Scoring Rules Minimization

**Amirhossein Vahidi**[a,b,c]  **Simon Schosser**[a]  **Lisa Wimmer**[a,b]  **Yawei Li**[a,b]  **Bernd Bischl**[a,b]
**Eyke Hüllermeier**[b,d]  **Mina Rezaei**[a,b]

[a]Department of Statistics LMU Munich, Munich, Germany
[b]Munich Center for Machine Learning, Munich, Germany
[c]Wellcome Sanger Institute, Cambridge, UK
[d]Institute of Informatics LMU Munich, Munich, Germany
{av13@sanger.ac.uk, mina.rezaei@stat.uni-muenchen.de}

## Abstract

Self-supervised learning methods have shown promising results across a wide range of tasks in computer vision, natural language processing, and multimodal analysis. However, self-supervised approaches come with a notable limitation, dimensional collapse, where a model doesn't fully utilize its capacity to encode information optimally. Motivated by this, we propose ProSMin, a novel probabilistic self-supervised learning approach that leverages the power of probabilistic models to enhance representation quality and mitigate collapsing representations. Our proposed approach involves two neural networks, the online network and the target network, which collaborate and learn the diverse distribution of representations from each other through probabilistic knowledge distillation. The two networks are trained via our new loss function based on proper scoring rules. We provide a theoretical justification for ProSMin and demonstrate its modified scoring rule. This insight validates the method's optimization process and contributes to its robustness and effectiveness in improving representation quality. We evaluate our probabilistic model on various downstream tasks, such as in-distribution generalization, out-of-distribution detection, dataset corruption, low-shot learning, and transfer learning. Our method achieves superior accuracy and calibration, outperforming the self-supervised baseline in a variety of experiments on large datasets such as ImageNet-O and ImageNet-C. ProSMin thus demonstrates its scalability and real-world applicability. Our code is publicly available: https://github.com/amirvhd/SSL-sore-rule.

## 1 Introduction

Self-supervised learning (SSL) is one of the most promising approaches for learning representations from a limited set of labeled data and has achieved outstanding results in several domains and applications, including natural language processing (NLP; Devlin et al. (2018); Brown et al. (2020); Saggau et al. (2023)), computer vision (Chen et al., 2020; Bardes et al., 2021; Grill et al., 2020; Rezaei et al., 2023a), multimodal learning (Radford et al., 2021; Li et al., 2022b; Shi et al., 2022), and bioinformatics (Gündüz et al., 2023; Rezaei et al., 2023b). However, *collapsing representations* is one of the significant drawbacks to current SSL methods, in which the representations converge to a limited set of points in the embedding space.

The dimensional collapse in SSL occurs due to excessive distortion caused by strong data augmentation, making the duplicates dissimilar to the originals collapsing certain dimensions, as well as network overparameterization which leads to lower-dimensional solutions (Jing et al., 2022). Strong augmentation can introduce more variance than present in the data distribution, causing collapse when the contrastive covariance matrix is not positive semidefinite, while overparameterization biases the network towards flatter minima, hindering information encoding in dimensions even for

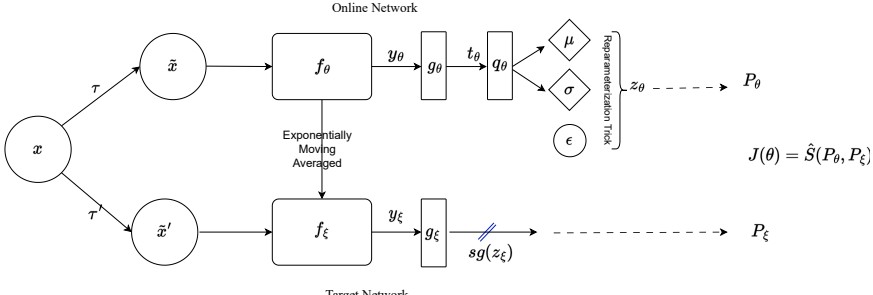

Figure 1: Illustration of our proposed probabilistic self-supervised learning via scoring rule minimization namely *ProSMin*. Given a batch $X$ of input samples, two different augmented samples $\tilde{x}$ and $\tilde{x}'$ are taken by an online network with $\theta$ and target network with $\xi$ parameters respectively. Our objective is to minimize the proposed scoring rule between $P_{\theta}$, $P_{\xi}$.

similar positive pairs. Understanding and addressing these mechanisms is crucial for enhancing the effectiveness of self-supervised learning and preventing undesirable dimensional collapses.

Collapsing representations lead to reduced representation quality, impaired generalization to new tasks or domains, and limited capacity to handle variations in the data (Jing et al., 2022). Most recent studies addressed this problem with contrastive learning through effective augmentation (Bai et al., 2022), negative sample strategies (Kalantidis et al., 2020), ensemble approaches (Vahidi et al., 2023; Ruan et al., 2022), regularization techniques (Li et al., 2022a; Rezaei et al., 2023a), removing correlations in the feature space (Bardes et al., 2022), clustering-based approaches (Assran et al., 2022; 2021) and self-distillation (Caron et al., 2021; Grill et al., 2020).

In this paper, we formulate the collapsing self-supervised representation problem through probabilistic machine learning, aiming to provide a comprehensive and nuanced solution that addresses the limitations of deterministic approaches. This integration allows us to offer predictive distributions alongside their associated point predictions (Ho et al., 2020; Nichol & Dhariwal, 2021). This pivotal shift offers promising avenues for achieving representation reliability and superior generalization capabilities in self-supervised learning scenarios.

Specifically, we propose ProSMin, a novel probabilistic self-supervised learning method that minimizes a scoring rule during pretext task learning. We motivate ProSMin by formulating knowledge distillation (KD) in a probabilistic manner. As shown in Figure 1, our proposed method involves two deep neural networks, an online and a target network, each learning a different representation of input samples. The online network maps input samples to a probability distribution inferred from the representation of the online encoder part. We train the online network in such a manner that samples from its output distribution predict the target network's representation on a second augmented view of the input. The loss realized by this prediction is expressed via a modified scoring rule, which incentivizes the recovery of the true distribution. Our contributions are:

- We introduce a novel probabilistic definition of robust representation for self-supervised learning. The probabilistic definition provides a deeper understanding of the quality and trustworthiness of the learned representations in guiding subsequent tasks.

- Our probabilistic approach effectively mitigates collapsing representations by encouraging the online and target networks to explore a diverse range of representations, thus avoiding convergence to a limited set, which results in a more comprehensive representation space that better encapsulates the intricacies of the data distribution.

- We discuss a rigorous theoretical foundation for our proposed algorithm. This theoretical insight not only underscores the robustness of our approach but also provides a principled explanation for its effectiveness in improving representation quality.

- Through extensive empirical analysis, we validate the effectiveness of our approach in diverse scenarios. Our method achieves competitive predictive performance and calibration on various tasks such as in-distribution (IND), out-of-distribution (OOD), and corrupted datasets, demonstrating generalization capabilities. Moreover, we demonstrate the superi-

ority of our method in semi-supervised and low-shot learning scenarios. Our framework establishes a superior trade-off between predictive performance and robustness when compared to deterministic baselines. This outcome is particularly notable on large-scale datasets such as ImageNet-O and ImageNet-C, underscoring the scalability and effectiveness of our method in real-world, high-dimensional settings.

## 2 BACKGROUND AND RELATED WORK

Self-supervised methods are designed to tackle unsupervised problems by training on a *pretext task* that utilizes the data itself to generate labels, effectively employing supervised methods to solve unsupervised problems (Grill et al., 2020; Chen et al., 2020; Jang et al., 2023; Caron et al., 2021; Zhou et al., 2021; Zbontar et al., 2021; Bardes et al., 2021; Chen et al., 2021). The resulting representations learned from the pretext task can serve as a foundation for *downstream supervised tasks*, such as image classification or object detection. Alternatively, the extracted representation can be directly utilized for downstream applications, such as detecting anomalies and OOD data (Tran et al., 2022). Recent studies (Oquab et al., 2023; Zhou et al., 2021) provided evidence that performing self-supervised pretext task learning on a large-scale and diverse dataset can extract features that are effective across different image distributions and tasks without the need for fine-tuning. Following this, we introduce a novel probabilistic self-supervised framework aiming to learn robust representation over parameters using *self-distillation* and by *minimizing a scoring rule*.

Self-distillation is a variant of knowledge distillation (Hinton et al., 2015) in which a larger model (*teacher*) is used to distill knowledge into a smaller model (*student*) of the same architecture (Caron et al., 2021; Zhou et al., 2021). Given an input sample $x$, the student network $f_\theta$ is trained on the soft labels provided by the teacher network $f_\xi$. Self-distillation combines self-supervised learning with knowledge distillation and was introduced by DINO (Caron et al., 2021). The two networks share the same architecture but take different augmentations of the input sample and output different representation vectors. The knowledge is distilled from the teacher network $f_\xi$ to student $f_\theta$ by minimizing the cross-entropy between the respective representation vectors. The parameters $\theta$ of the student network are obtained from an exponential moving average (EMA) of the parameters $\xi$ of the teacher network, thus reducing computational cost by confining backpropagation to the teacher network. In this paper, we present a novel approach to knowledge distillation aimed at training a network in a probabilistic manner using parametrization trick (Kingma et al., 2015) and a novel scoring rule objective function.

A scoring rule is a function used to evaluate the accuracy of a probabilistic prediction (Gneiting & Raftery, 2007; Der Kiureghian & Ditlevsen, 2009; Hanselle et al., 2023). It quantifies the divergence between the predicted probability distribution and the true distribution of the event. The concept of a scoring rule is fundamental to many areas of machine learning, including probabilistic classification (Parry, 2016) and decision theory (Dawid & Musio, 2014). A proper scoring rule (Gneiting & Katzfuss, 2014) is one that incentivizes truthful reporting of the probabilities by the forecaster, i.e., the forecaster is incentivized to report the correct probability distribution (Pacchiardi et al., 2021). The use of proper scoring rules has had significant implications in areas such as online learning (V'yugin & Trunov, 2019), generative neural networks (Pacchiardi & Dutta, 2022), and uncertainty quantification (Bengs et al., 2023; Gruber & Buettner, 2022; Sale et al., 2023). This paper utilizes scoring rules and adapts them for the purpose of pretext task learning of a self-supervised framework. This adaptation is inspired by the endeavor to infuse probabilistic learning principles into the self-supervised learning domain.

## 3 PROBLEM FORMULATION

**Avoiding collapsing representation** One of the major challenges in self-supervised learning is collapsing representations, where learned representations converge to a limited set of points in the representation space. In other words, the model fails to capture the full diversity and richness of the underlying data distribution. This can lead to reduced representation quality, impaired generalization to new tasks or domains, and limited capacity to handle variations in the data. various techniques have been explored to mitigate dimensional collapse including contrastive losses (Kalantidis et al., 2020) that separate embedding of negative examples, non-contrastive losses (Rezaei et al., 2023a; Bardes et al., 2022) that reduce informational redundancy among embeddings, and joint embedding

and clustering approaches (Assran et al., 2022; 2023) aiming to increase the average embedding entropy. In this paper, we formulate the collapsing representation problem through a probabilistic lens, aiming to provide a comprehensive and nuanced solution that not only addresses the limitations of deterministic approaches but also harnesses the power of uncertainty quantification and broader representation distributions.

**Scoring rules**  A scoring rule (Gneiting & Raftery, 2007) is a function that evaluates how well a predicted distribution $P$ over a random variable $\mathbf{X}$ aligns with the actually observed realizations $\boldsymbol{x}$ of $\mathbf{X}$. We define the loss[1] of predicting distribution $P$ while observing $\boldsymbol{x}$ as $S(P, \boldsymbol{x})$. Assuming that $\mathbf{X}$ follows some true distribution $Q$, the *expected* scoring rule measuring the loss of predicting $P$ can be expressed as $S(P, Q) \triangleq \mathbb{E}_{\mathbf{X} \sim Q} S(P, \mathbf{X})$. A scoring rule $S$ is *proper* with respect to a set of distributions $\mathcal{P}$ if for all $P, Q \in \mathcal{P}$ it holds that $S(Q, Q) \leq S(P, Q)$, thus incentivizing the prediction of the true distribution $Q$. If the former holds with equality, i.e., the expected score $S(P, Q)$ is uniquely minimized in $Q$ at $Q = P$, then the scoring rule is called *strictly proper*. In practice, the expectation with respect to $Q$ is usually replaced by an empirical mean over a finite amount of samples. We refer to the resulting scoring rule as $\hat{S}$. There are many types of scoring rules, including entire parameterized families of strictly proper scoring rules. In Appendix 10, we provide more details about some of the rules we use in our experiments.

## 4 METHOD

Consider a randomly sampled mini-batch of training data $\boldsymbol{X} \triangleq [\boldsymbol{x}_1, \ldots, \boldsymbol{x}_n]$ where $\boldsymbol{x}_i \in \mathbb{R}^D$ and transformation functions $\tau, \tau'$ acting on the data. To enhance the training process, the transformation functions produce two augmented views $\tilde{\boldsymbol{x}} \triangleq \tau(\boldsymbol{x})$ and $\tilde{\boldsymbol{x}}' \triangleq \tau'(\boldsymbol{x})$ for each sample in $\boldsymbol{X}$. These augmented views are generated by sampling $\tau, \tau'$ from the distribution of suitable data transformations, such as partially masking image patches (He et al., 2022) or applying image augmentation techniques (Chen et al., 2020).

As depicted in Fig. 1, the first augmented view $\tilde{\boldsymbol{x}}$ is fed to the encoder of online network $f_{\boldsymbol{\theta}}$ that outputs a *representation* $y_{\boldsymbol{\theta}} \triangleq f_{\boldsymbol{\theta}}(\tilde{\boldsymbol{x}})$. This is followed by passing it subsequently through a projector $g_{\boldsymbol{\theta}}$ and predictor $q_{\boldsymbol{\theta}}$, such that $t_{\boldsymbol{\theta}} \triangleq g_{\boldsymbol{\theta}}(y_{\boldsymbol{\theta}})$ and $z_{\boldsymbol{\theta}} \triangleq q_{\boldsymbol{\theta}}(t_{\boldsymbol{\theta}})$. We collect all trainable parameters of the online network in $\boldsymbol{\theta}$. Similarly, the encoder of target network $f_{\boldsymbol{\xi}}$ takes the second augmented view and outputs $y_{\boldsymbol{\xi}} \triangleq f_{\boldsymbol{\xi}}(\tilde{\boldsymbol{x}}')$, followed by the projector network $g_{\boldsymbol{\xi}}$ producing $t_{\boldsymbol{\xi}} \triangleq g_{\boldsymbol{\xi}}(y_{\boldsymbol{\xi}})$, where the trainable parameters are denoted as $\boldsymbol{\xi}$. It is important to note that the predictor is applied exclusively to the online network.

In order to introduce probabilistic self-distillation training, we employ a scoring rule (Gneiting & Raftery, 2007) as our loss function. To accomplish this, it is necessary to generate samples from the online network. However, directly sampling $\mu$ and $\sigma$ from a deterministic vector $q_{\boldsymbol{\theta}}(t)$ is inadequate, as it prevents us from performing backpropagation effectively. One way of producing samples from a neural network architecture, while still enabling backpropagation with respect to the latent representation $\boldsymbol{z}$, is to use the reparametrization trick (Kingma et al., 2015). Here, we assume that the output of the predictor network $z_{\boldsymbol{\theta}} = q_{\boldsymbol{\theta}}(t_{\boldsymbol{\theta}})$ follows an underlying normal distribution with mean $\mu$ and standard deviation $\sigma$. We generate $r \in \mathbb{N}$ samples from the output of the linear layers following the prediction head by sampling random noise $\epsilon_j^i \sim N(0, 1)$ for each augmented view of the $i$-th data point, such that the $j$-th sample, with $j \in 1, ..., r$, is given by: $\boldsymbol{z}_j^i = \mu^i + \sigma^i \odot \epsilon_j^i$.

Thus, we obtain samples $\boldsymbol{z}_j^i$ by shifting and scaling the random noise samples $\epsilon_j^i$ by the outputs $(\mu, \sigma)$ of a neural network with trainable parameters $\boldsymbol{\theta}$, and the loss incurred by these samples can be backpropagated to update $\boldsymbol{\theta}$ during training. The online network parameters are updated by minimizing the scoring rule as follows:

$$\hat{\boldsymbol{\theta}} := \arg\min_{\boldsymbol{\theta}} J(\boldsymbol{\theta}), \quad J(\boldsymbol{\theta}) = S(P_{\boldsymbol{\theta}}, P_{\boldsymbol{\xi}}) := \mathbb{E}_{z_{\boldsymbol{\xi}} \sim P_{\boldsymbol{\xi}}}[P_{\boldsymbol{\theta}}, z_{\boldsymbol{\xi}}], \tag{1}$$

---

[1]The original proposal in Gneiting & Raftery (2007) defines scoring rules in terms of a gain that is to be maximized. We adhere to the convention in deep learning of expressing the objective via a loss function that we seek to minimize.

where $P_{\boldsymbol{\xi}}$ denotes the target distribution and $z_{\boldsymbol{\xi}}$ denotes the target output and $P_{\boldsymbol{\theta}}$ represents the online induced multivariate normal distribution. We define the approximation of our customized scoring rule loss for $P_{\boldsymbol{\theta}}$ and $P_{\boldsymbol{\xi}}$ as follows:

$$\hat{S}(P_{\boldsymbol{\theta}}, P_{\boldsymbol{\xi}}) = \frac{1}{N} \sum_{i=1}^{N} \left[ \frac{2\lambda}{r} \sum_{j=1}^{r} \|\boldsymbol{z}_j^i - \boldsymbol{z}_\xi^i\|_2^\beta - \frac{1-\lambda}{r(r-1)} \sum_{j \neq k} \|\boldsymbol{z}_j^i - \boldsymbol{z}_k^i\|_2^\beta \right] \tag{2}$$

where $\boldsymbol{z}_\xi^i$ represents the target prediction for the $i$-the input sample. $\beta \in (0, 2)$ and $\lambda \in (0, 1)$ are learn-able hyperparameters. We set $\lambda$ so that $\hat{S}(P_{\boldsymbol{\theta}}, z_{\boldsymbol{\xi}}) > 0$. See Appendix 10.5.6 for detailed information.

By the principle of knowledge distillation, the parameters of the target network are updated through the EMA of the weights from the online network (Grill et al., 2020; Caron et al., 2021), saving the need for backpropagation and thus reducing computation time considerably.

$$\boldsymbol{\xi}_t = (1-\alpha)\boldsymbol{\theta}_t + \alpha\boldsymbol{\xi}_{t-1}, \quad t = 1, 2, \ldots, \quad \alpha \in [0, 1] \tag{3}$$

The initial weights $\boldsymbol{\xi}_0$ are obtained through random initialization, while $\alpha$ is a hyperparameter that determines the updating rate of the target weights and is set to change from 0.9 to 1 during training with a cosine scheduler.

Next, we provide a comprehensive description of our objective function based on scoring rules (Section 4.1). For a detailed understanding of scoring rules and their various variants, please refer to Section 10.

## 4.1 OBJECTIVE FUNCTION

Numerous scoring rules can be decomposed into two terms. The first term is a function of $\boldsymbol{z}_j$ and the realized observation $\boldsymbol{z}_\xi$. The second term is a function of two samples, $\boldsymbol{z}_j$ and $\boldsymbol{z}_k$, drawn from the predicted distribution $P$. Our objective function is an adjusted version of the former, where the two components are posed as a convex combination with component weights controlled via hyperparameter $\lambda \in (0, 1)$. We set $\lambda$ so that $\hat{S}(P_{\boldsymbol{\theta}}, z_{\boldsymbol{\xi}}) > 0$. This modification of the scoring rule can be useful to adjust the focus of the loss function on either part of the scoring rule. Two notable examples of scoring rules adhering to this form are the *energy score* and the *kernel score* (Gneiting & Raftery, 2007).

With the above notation, we define the energy score as:

$$S_{\mathrm{E}}(P_{\boldsymbol{\theta}}, \boldsymbol{z}_{\boldsymbol{\xi}}) = 2 \cdot \mathbb{E}_{P_{\boldsymbol{\theta}}} \left[ \|\boldsymbol{z}_j - \boldsymbol{z}_{\boldsymbol{\xi}}\|_2^\beta \right] + \left( -\mathbb{E}_{P_{\boldsymbol{\theta}}} \left[ \|\boldsymbol{z}_j - \boldsymbol{z}_k\|_2^\beta \right] \right) =: S_{\mathrm{E}}^1(P_{\boldsymbol{\theta}}, \boldsymbol{z}_{\boldsymbol{\xi}}) + S_{\mathrm{E}}^2(P_{\boldsymbol{\theta}}) \tag{4}$$

Analogously, we write the kernel score as:

$$S_{\mathrm{K}}(P_{\boldsymbol{\theta}}, \boldsymbol{z}_{\boldsymbol{\xi}}) = \mathbb{E}_{P_{\boldsymbol{\theta}}} \left[ k(\boldsymbol{z}_j, \boldsymbol{z}_k) \right] + \left( -2 \cdot \mathbb{E}_{P_{\boldsymbol{\theta}}} \left[ k(\boldsymbol{z}_j, \boldsymbol{z}_{\boldsymbol{\xi}}) \right] \right) =: S_{\mathrm{K}}^2(P_{\boldsymbol{\theta}}) + S_{\mathrm{K}}^1(P_{\boldsymbol{\theta}}, \boldsymbol{z}_{\boldsymbol{\xi}}), \tag{5}$$

with suitable kernel function $k(\cdot, \cdot)$.

For simplicity, we only write $S(P_{\boldsymbol{\theta}}, \boldsymbol{z}_{\boldsymbol{\xi}}) = S^1(P_{\boldsymbol{\theta}}, \boldsymbol{z}_{\boldsymbol{\xi}}) + S^2(P_{\boldsymbol{\theta}})$ in the following for both scores. With $\lambda \in (0, 1)$, we define the general form of our objective function as follows:

$$S^*(P_{\boldsymbol{\theta}}, \boldsymbol{z}_{\boldsymbol{\xi}}) := \lambda S^1(P_{\boldsymbol{\theta}}, \boldsymbol{z}_{\boldsymbol{\xi}}) + (1-\lambda)S^2(P_{\boldsymbol{\theta}}) \tag{6}$$

## 4.2 THEORETICAL JUSTIFICATION

Our approach is motivated by a strictly proper scoring rule, which ensures that the expected score is optimized only by the true distribution. However, our modified scoring rule gives more weight to the online network predictions during training. This deliberate focus is aimed at enhancing the model's learning efficacy. By optimizing the relative importance of each component using a hyperparameter $\lambda$, we encourage the model not only to fit the data but also to actively refine its predictive capabilities. Our proposed loss function (Eq. 6) consists of two key elements:

- Comparison with the target network ($S^1$): This term measures the adherence of the model's predictions to a known benchmark, ensuring that the model remains grounded in established knowledge.

- Entropy for online network predictions ($S^2$): This term emphasizes the internal consistency and reliability of the model's own predictions over time. This term increases the entropy of prediction and helps generalization by helping the exploration power of the model.

In its original form, the scoring rule can exhibit a slight bias toward matching the target network too closely. To address this, we introduce stochasticity into the training process by increasing the weight of the second term (comparing online network predictions) in the loss function. This is not an artifact of poor design but a strategic insertion of complexity that nudges the algorithm towards more sophisticated regions of the solution space, thereby potentially locating a better optimum. Empirically, we show that an "improper" scoring rule variant can paradoxically increase accuracy.

## 5 IMPLEMENTATION DETAILS AND EXPERIMENTAL SETUP

**Image augmentation** We define a random transformation function $T$ that applies a combination of multi-crop, horizontal flip, color jittering, and grayscale. Similar to Caron et al. (2021), we perform multi-crops with a random size from $0.8$ to $1.0$ of the original area and a random aspect ratio from $3/4$ to $4/3$ of the original aspect ratio. We define color-jittering of $(0.8, 0.8, 0.8, 0.2)$, and Gaussian blurring with $0.5$ probability and $\zeta = (0.1, 2.0)$.

**Deep self-supervised network architecture** The *online neural network* is constructed from a backbone $f$, which can be either ViT (Dosovitskiy et al., 2021) or ResNet (He et al., 2016), and a projection head $g$ followed by a prediction $q$. The backbone output $f$ is used as a feature for downstream tasks. The projection consists of a 3-layer multilayer perceptron (MLP) with a hidden dimension of 2048, followed by 2 normalizations and a weight-normalized fully connected layer with $K$ dimensions, similar to the design used in the DINO projection head. We use a predictor with two layers of MLPs with a hidden dimension of 12000, with a GELU nonlinearity in between. Note that ViT architectures do not use batch normalization (BN) by default, so we do not use BN in the projection or prediction when using ViT.

The *target network* has the same backbone and projection as the online network and the target network learns through self-distillation. Similar to DINO Caron et al. (2021), after the online network parameters are updated, an EMA of the online parameters (i.e., a momentum encoder) is used to update the target parameters. The EMA prevents the target parameters from being updated too quickly. After the parameters are updated, the target also receives a new centering parameter.

**Optimization** Our pretraining process involves training the models on the ImageNet training dataset (Deng et al., 2009) using the *adamw* optimizer (Loshchilov & Hutter, 2017a) and a batch size of 512, distributed across 8 GPUs using Nvidia Tesla A100 with ViT-S/16 architecture. We adopt a linear scaling rule to determine the base value of the learning rate, which is ramped up linearly during the first 30 epochs. Specifically, the learning rate is set to $lr = 0.0005 * \text{batchsize}/256$. After the warmup phase, we decay the learning rate using a cosine schedule (Loshchilov & Hutter, 2017b). The weight decay also follows a cosine schedule, increasing from $0.04$ to $0.4$. The centering (smooth parameter) is $0.9$.

**Datasets** The datasets utilized in our experiments are as follows: The **ImageNet** (Deng et al., 2009) dataset with 1.28 million training images and 50,000 validation images with the size of $256 \times 256$ contains 1,000 classes. The **ImageNet-O** dataset (Srivastava et al., 2022) comprises images belonging to classes that are not present in the ImageNet-1k dataset. It is considered a challenging benchmark for evaluating the robustness of the models, as it requires models to generalize to a diverse range of visual conditions and handle variations that are not typically encountered in standard training datasets. The **ImageNet-C** (Hendrycks & Dietterich, 2019) is a benchmark for evaluating the robustness of the models against common corruptions and perturbations that can occur in real-world scenarios. It consists of more than 30,000 images derived from the ImageNet dataset, with each image being corrupted in one of 15 different ways, including noise, blur, weather conditions, and digital artifacts. **CIFAR-10/100** (Krizhevsky, 2009) are subsets of the tiny images dataset. Both datasets include 50,000 images for training and 10,000 validation images of size $32 \times 32$ with 10 and 100 classes, respectively. The **Oxford 102 Flower** (Nilsback & Zisserman, 2008) consists of 102 flower categories, each class including between 40 and 258 images. The images have large scale, pose, and light variations. In addition, there are categories that have large variations within the category

and several very similar categories. **iNaturalist-2018** (Van Horn et al., 2018) (iNat) comprises a vast collection of 675,170 training and validation images, classified into 5,089 distinct fine-grained categories found in the natural world. It is worth noting that the iNat dataset exhibits a significant imbalance, as the number of images varies greatly across different categories.

**Tasks**   We evaluate the performance of ProSMin representations after self-supervised pretraining on the ImageNet on the basis of **In-Domain (IND) generalization**, **OOD detection**, **semi-supervised learning**, **low-shot learning**, **corrupted dataset evaluation** (see Section 6), as well as **transfer learning to other datasets and tasks** (see Section 11).

**Evaluation metrics**   We report the prediction performance with the following metrics: **Top-1 accuracy** $\uparrow$: refers to the proportion of test observations that are correctly predicted by the model's output as belonging to the correct class. **AUROC** $\uparrow$: the area under the ROC curve represents the relationship between false-positive and false-negative rates for various classification thresholds. In this case, the positive and negative classes refer to whether an observation is in or out of a given distribution, respectively, and the ROC curve is plotted as the threshold for classifying an observation as "positive" is gradually increased. **Negative log-likelihood (NLL)** $\downarrow$: measures the probability of observing the given test data given the estimated model parameters, multiplied by -1. This measure quantifies the degree to which the model's estimated parameters fit the test observations. **Expected calibration error (ECE)** $\downarrow$ (Naeini et al., 2015): calculated as the mean absolute difference between the accuracy and confidence of the model's predictions, where confidence is defined as the highest posterior probability among the predicted classes. The difference is calculated across equally-spaced confidence intervals or bins and is weighted by the relative number of samples in each bin. A lower value of ECE indicates better calibration of the model. **Mean Calibration Error (mCE)** $\downarrow$ is a metric used to evaluate the calibration of a classification model, similar to ECE. It is calculated as the mean of the absolute differences between the predicted and true probabilities of a given class across all classes. A lower value of mCE indicates better calibration of the model.

## 6   RESULTS AND DISCUSSION

**In-distribution generalization**   IND generalization (or *linear evaluation*) measures how well a model's confidence aligns with its accuracy. To assess and compare the predictive abilities of our proposed model on in-distribution datasets, we freeze the encoder of the online network, denoted as $f_\theta$, after performing unsupervised pretraining. Then, we train a supervised linear classifier using a fully connected layer followed by softmax, which is placed on top of $f_\theta$ after removing the projection and prediction network. The desired outcome is high predictive scores and low uncertainty scores. In Table 1, a comprehensive comparison is presented between our approach and other self-supervised methods. The results indicate that our method outperforms all others in terms of top-1 accuracy, and calibration, as demonstrated by the lowest expected calibration error (ECE) and negative log-likelihood (NLL) scores.

**Out-of-distribution detection**   The ability of a model to recognize test samples from classes that were not present during training is evaluated using OOD detection, as discussed in Geng et al. (2020). We conduct experiments on ImageNet-O (Srivastava et al., 2022) to assess the generalization of the model from IND to OOD datasets, as well as to predict the uncertainty of the models on OOD datasets. Note that evaluation is performed directly after self-supervised pretraining without a fine-tuning step. The y-axis of Figure 2 shows and compares the results for the OOD task in terms of AUROC. Remarkably, our method shows outstanding results for the detection of OOD samples compared to other approaches such as i-BOT, DINO, and Moco-V3. This outcome aligns with our expectations as we directly use the probabilistic latent representation for this task.

**Corrupted dataset evaluation**   An essential aspect of model robustness is its capability to produce precise predictions when the test data distribution changes. We examine model robustness under the context of *covariate shift*. Fig. 2 presents the improved performance metrics. Our method outperforms the baseline and has comparable predictive performance to the baseline for mCE.

**Semi-supervised and low-shot learning on ImageNet**   Following the semi-supervised protocol established in Chen et al. (2020), we employ fixed 1% and 10% splits of labeled training data from ImageNet. In Table 2, we compare our performance against several concurrent models, including the baseline (DINO). Based on results obtained in Table 2, our approach outperforms state-of-the-art methods in semi-supervised evaluation for both 1% and 10% scenarios. We assess our model's efficacy on a low-shot image classification task, where we train logistic regression on frozen weights

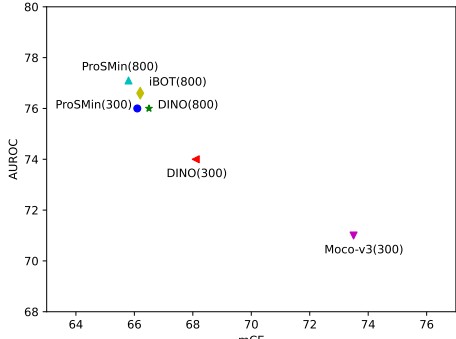

Figure 2: **OOD detection and corrupted dataset evaluation**. Methods with higher AUROC and lower mCE are better. Our method demonstrates outstanding performance among other approaches. We present and compare the performance of our method with others on the task of OOD detection on ImageNet-O and corrupted datasets on ImageNet-C, in Table 5

with 1% and 10% labels. It's important to note that this experiment was performed on frozen weights, without finetuning. Table 2 shows our features achieved better performance compared to state-of-the-art methods.

Table 1: **IND Generalization (or Linear Evaluation)**. Top-1 accuracy, $\kappa$-NN, ECE, and NLL averaged over in-distribution on test samples of the **ImageNet** dataset where the encoder is *ViT-S/16* over 800 epochs. The best score for each metric is shown in **bold**, and the second-best is underlined.

| Method | Top-1 Acc (%) ($\uparrow$) | $\kappa$-NN ($\uparrow$) | NLL ($\downarrow$) | ECE ($\downarrow$) |
|---|---|---|---|---|
| DINO Caron et al. (2021) | 76.8 | 74.5 | 0.919 | 0.015 |
| MOCO-V3 Chen et al. (2021) | 73.2 | 64.7 | 1.152 | 0.027 |
| i-BOT Zhou et al. (2021) | 77.9 | 75.2 | 0.918 | 0.013 |
| ProSMin | **78.4** | **76.2** | **0.900** | **0.006** |

# 7 ABLATION STUDY

To gain a deeper understanding of the behavior and performance of our proposed method, we conducted several ablation studies to explore various aspects of our approach. Specifically, we investigate the following factors: different scoring rules as an objective function, the hyperparameter of our loss function ($\lambda$), the number of samples used for generating latent representations, the dimension of the embedding space, the effect of the momentum hyperparameter, the impact of batch normalization (BN), and the prediction layer (PL). We also supply an ablation analysis on computational efficiency and the broader impact of our method in the Appendix (see Section 11). These investigations aim to provide insights and intuition regarding our approach.

**Impact of different components of scoring rule** We conduct a series of experiments to explore alternative scoring rules, including kernel scoring rules and various variations of energy scoring rules, for our objective function. The results, as presented in Table 3, demonstrate that the kernel scoring

Table 2: **Low-shot and semi-supervised evaluation**: Top-1 accuracy (ACC), ECE, and NLL for semi-supervised on ImageNet classification using 1% and 10% training examples fine-tuning. and Low-shot results with frozen ViT features.

| Method | 1% | 10% | Architecture | Parameters |
|---|---|---|---|---|
| Semi-supervised | | | | |
| DINO Caron et al. (2021) | 60.3 | 74.3 | ViT-S/16 | 21 |
| i-BOT Zhou et al. (2021) | 61.9 | 75.1 | ViT-S/16 | 21 |
| ProSMin (ours) | **62.1** | **75.6** | ViT-S/16 | 21 |
| Low-shot learning | | | | |
| DINO Caron et al. (2021) | 64.5 | 72.2 | ViT-S/16 | 21 |
| i-BOT Zhou et al. (2021) | 65.9 | 73.4 | ViT-S/16 | 21 |
| ProSMin (ours) | **66.1** | **73.8** | ViT-S/16 | 21 |

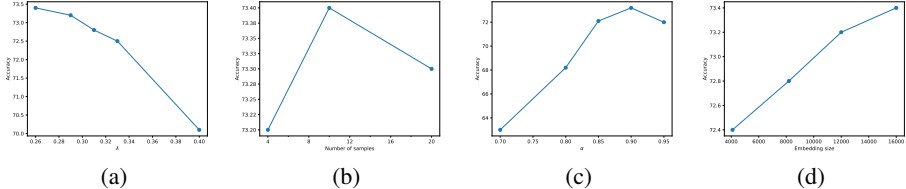

|     (a)     |     (b)     |     (c)     |     (d)     |

Figure 3: Study of hyperparameters of our proposed ProSMin (a) $\lambda$, (b) Number of samples, (c) Momentum coefficient ($\alpha$), and (d) Size of embedding obtained by 100 epochs on ImageNet.

Table 3: **Important component for training**. We investigate the effect of each component on the linear evaluation performance for 100 epochs. The first line shows the best combination. PL is the prediction layer, BN is the batch normalization layer.

| Energy (L1 loss) | Kernel score | Energy (L2 loss) | BN. | PL | $\lambda$ | $\sigma$ | Accuracy |
|---|---|---|---|---|---|---|---|
| ✓ | ✗ | ✗ | ✗ | ✓ | ✓ | ✓ | 73.2 |
| ✗ | ✓ | ✗ | ✗ | ✓ | ✓ | ✓ | 1.1 |
| ✗ | ✗ | ✓ | ✗ | ✓ | ✗ | ✗ | 43.0 |
| ✓ | ✗ | ✗ | ✓ | ✓ | ✓ | ✓ | 68.5 |
| ✓ | ✗ | ✗ | ✗ | ✗ | ✓ | ✓ | 71.2 |
| ✓ | ✗ | ✗ | ✗ | ✓ | ✗ | ✗ | 47.8 |

rule exhibits instability. Furthermore, we investigate a case with L1 loss (the last row of Table 3) with prediction layer and without batch normalization,that the $\lambda$ and $\sigma$ are zero, to motivate the effect of the second part of our loss function. Overall, the energy score with $\beta = 1$ ($L1$ loss) yields the best performance among the tested scoring rules. We provide a theoretical explanation for $L1$ in Section 10.5.3.

**Study of hyperparameters** Figure 3a depicts the influence of the hyperparameter $\lambda$ utilized in our proposed loss function, which controls the impact of $S^1(P_{\boldsymbol{\theta}}, \boldsymbol{z_\xi})$ and $S^2(P_{\boldsymbol{\theta}}, \boldsymbol{z_\xi})$ on the objective function. An ablation analysis was conducted to investigate the effect of increasing the number of samples from $q_{\boldsymbol{\theta}}$, as illustrated in Figure 3b. The results demonstrate that employing four samples yields satisfactory performance. Figure 3c showcases the outcomes obtained from the knowledge distillation rate. In previous approaches, the exponential moving average parameter initiated from a value relatively close to 1 (e.g., 0.996 (Grill et al., 2020), Caron et al. (2021)). However, in our case, $\alpha$ starts from 0.9, implying a faster pace of knowledge distillation. Furthermore, we examine the impact of different sizes for the embedding vector, as presented in Fig. 3d. The results obtained after 100 epochs reveal that increasing the embedding size leads to improved performance. However, it should be noted that larger embedding sizes necessitate additional computational resources, thus our choice of size is based on the available computational capacity. Table 3 provides insights into the influence of batch normalization in the prevention of representation collapse (Grill et al., 2020). As our framework operates in a probabilistic setting, the inclusion of batch normalization is unnecessary for averting collapse. Additionally, the prediction layer ($q_{\boldsymbol{\theta}}$) enhances performance by facilitating improved feature extraction in online networks.

# 8 CONCLUSION

In this paper, we presented *ProSMin* as a novel probabilistic self-supervised framework to address a collapsing representation problem. Our method includes two neural networks that collaborate and learn from each other using an augmented format. Our framework is trained by minimizing a proposed scoring rule objective function. We evaluated ProSMin across different tasks, including in-distribution generalization, out-of-distribution detection, dataset corruption, transfer learning, low-shot, and semi-supervised learning. The results demonstrate that our method achieves superior performance in terms of accuracy and calibration, thus showing the effectiveness of our proposed approach.

## 9 ACKNOWLEDGMENT

M. R. and B. B. were supported by the Bavarian Ministry of Economic Affairs, Regional Development and Energy through the Center for Analytics – Data – Applications (ADA-Center) within the framework of BAYERN DIGITAL II (20-3410-2-9-8).M. R., B. B., E. H., L. W.,Y. L., and A.V. were supported by the German Federal Ministry of Education and Research (BMBF) Munich Center for Machine Learning (MCML). L. W. was supported by the DAAD program Konrad Zuse Schools of Excellence in Artificial Intelligence, sponsored by the German Federal Ministry of Education and Research

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

## 10 SCORING RULES

The notation employed in this study aligns with the work of Gneiting & Raftery (2007) regarding scoring rules $S(P, \boldsymbol{x})$, where $P$ represents the predictive distribution and $\boldsymbol{x}$ denotes the observed data. By assuming that $\mathbf{X}$ follows a true distribution $Q$. With this assumption, we first explain different variations of scoring rules, such as the expected scoring rule (10.1), the proper scoring rule (10.2), the energy scoring rule (10.3), and the kernel scoring rule (10.4).

We then discuss the unbiased estimation properties of the scoring rule 10.5 for kernel score 10.5.1 and energy score 10.5.2, which we used as the objective function in our study.

In subsection 10.5.4, we establish the interchangeability of the expectation and gradient in our theoretical derivative, enabling us to derive gradients in a conventional manner. This is crucial due to the incorporation of samples within the scoring rule and the utilization of non-differential activation functions. Subsequently, in subsection 10.5.5, we introduce an unbiased estimate of the gradient.

### 10.1 EXPECTED SCORING RULE

We define the scoring rule $S(P, x)$ as a function of the distribution $P$ and the observation $x$ of $\mathbf{X}$. The expected scoring rule is defined as:

$$S(P, Q) := \mathbb{E}_{\mathbf{X} \sim Q} S(P, \mathbf{X}). \tag{7}$$

### 10.2 PROPER SCORING RULE

A scoring rule $S$ is called *proper* w.r.t a set of distributions $\mathcal{P}$, if for all $P, Q \in \mathcal{P}$ the expected score $S(P, Q)$ is minimized in $Q$ at $Q = P$. A scoring rule $S$ is called *strictly proper* if there exists the unique minimum $S(P, Q) > S(Q, Q)$ for $Q \neq P$.

### 10.3 ENERGY SCORE

The energy score is defined by:

$$S_{\mathrm{E}}^{(\beta)}(P, \mathbf{x}) = 2 \cdot \mathbb{E}\left[\|\tilde{\mathbf{X}} - \mathbf{x}\|_2^\beta\right] - \mathbb{E}\left[\left\|\tilde{\mathbf{X}} - \tilde{\mathbf{X}}'\right\|_2^\beta\right], \quad \tilde{\mathbf{X}} \perp \tilde{\mathbf{X}}' \sim P, \tag{8}$$

where $\beta \in (0, 2)$.

It is a strictly proper scoring rule for the class of probability measures $\mathcal{P}$ such that $\mathbb{E}_{\tilde{\mathbf{X}} \sim P}\|\tilde{\mathbf{X}}\|^\beta < \infty$. An unbiased estimate can be obtained by replacing the expectations in $S_{\mathrm{E}}^{(\beta)}$ with empirical means over draws from $P$.

### 10.4 KERNEL SCORE

Let $k(\cdot, \cdot)$ be a positive definite kernel. The kernel score for $k$ is defined by

$$S_k(P, \mathbf{x}) = \mathbb{E}\left[k\left(\tilde{\mathbf{X}}, \tilde{\mathbf{X}}'\right)\right] - 2 \cdot \mathbb{E}[k(\tilde{\mathbf{X}}, \mathbf{x})], \quad \tilde{\mathbf{X}} \perp \tilde{\mathbf{X}}' \sim P. \tag{9}$$

The kernel score is a proper scoring rule for the class of probability distribution $P$ for which holds, that $\mathbb{E}_{\tilde{\mathbf{X}}, \tilde{\mathbf{X}}' \sim P}\left[k\left(\tilde{\mathbf{X}}, \tilde{\mathbf{X}}'\right)\right] < \infty$. Under the condition that the kernel Maximum Mean Discrepancy is a metric, the kernel score is strictly proper. This condition is satisfied by the Gaussian kernel, which will be used in this work. Let $\gamma$ be a scalar bandwidth. We define the Gaussian kernel as follows:

$$k(\tilde{\mathbf{x}}, \mathbf{x}) = \exp\left(-\frac{\|\tilde{\mathbf{x}} - \mathbf{x}\|_2^2}{2\gamma^2}\right).$$

We get an unbiased estimate by replacing the expectations in $S_{\mathrm{k}}$ with the empirical means overdraws from $P$.

## 10.5 Unbiased estimators of the expected scoring rule

Let $N$ be the batch size and $r$ be the number of samples drawn from the online-induced distribution. Let $z_\theta = (z_1, ..., z_r)$ be the sample drawn from the online. We recall the definition of the Expected Scoring Rule:

$$S(P_\theta, P_\xi) := \mathbb{E}_{z_\xi \sim P_\xi}[S(P_\theta, z_\xi)],$$

where $z_\xi$ denotes the output of the target network while $P_\theta$ represents the online induced multivariate normal distribution.

### 10.5.1 Kernel score

We can write the estimated kernel score as follows:

$$\hat{S}(P_\theta, P_\xi) = \frac{1}{N} \sum_{i=1}^{N} \left[ \frac{1}{r(r-1)} \sum_{j,k=1, j\neq k} k(z_j^i, z_k^i) - \frac{2}{r} \sum_{j=1}^{r} k(z_j^i, z_\xi^i) \right],$$

### 10.5.2 Energy score

Consider $\beta \in (0, 2)$ and $N$ be the batch size. The Estimated Energy Score can be written as:

$$\hat{S}(P_\theta, P_\xi) = \frac{1}{N} \sum_{i=1}^{N} \left[ \frac{2}{r} \sum_{j=1}^{r} \|z_j^i - z_\xi^i\|_2^\beta - \frac{1}{r(r-1)} \sum_{j,k=1, j\neq k} \|z_j^i - z_k^i\|_2^\beta \right]$$

### 10.5.3 Partial derivatives of the estimated energy score

Let $\beta \in (0, 2)$ and let $N$ be the batch size. Let $x_i$ be the input data for $i \in 1, ..., N$. Let $r \in \mathbb{N}$ be the number of samples. Let $\mu_i = f_\theta^{(1)}(x_i)$ and let $\sigma_i^2 = f_\theta^{(2)}(x_i)$, where $f^{(j)}$ denotes the separated branches for mean and variance in the last layer of the online network for $j \in 1, 2$. We can calculate the partial derivatives needed for stochastic gradient descent as follows. Then, the formula for the mean is:

$$\frac{\partial}{\partial \mu} \hat{S}(P_\theta, P_\xi) = \frac{\partial}{\partial \mu} \frac{1}{N} \sum_{i=1}^{N} \left[ \frac{2}{r} \sum_{j=1}^{r} \left\| \mu_i + \epsilon_j \sqrt{\sigma_i^2} - \phi_\xi^i \right\|_2^\beta \right.$$

$$\left. - \frac{1}{r(r-1)} \sum_{j,k=1, j\neq k} \left\| \mu_i + \epsilon_j \sqrt{\sigma_i^2} - \mu_i + \epsilon_k \sqrt{\sigma_i^2} \right\|_2^\beta \right]$$

$$= \frac{1}{N} \sum_{i=1}^{N} \frac{2}{r} \sum_{j=1}^{r} \beta \left\| \mu_i + \epsilon_j \sqrt{\sigma_i^2} - \phi_\xi^i \right\|_2^{\beta-1} \frac{\partial}{\partial \mu} \left[ \mu_i + \epsilon_j \sqrt{\sigma_i^2} - \phi_\xi^i \right]$$

$$= \frac{1}{N} \sum_{i=1}^{N} \frac{2}{r} \sum_{j=1}^{r} \beta \left\| f_\theta^{(1)}(x_i) + \epsilon_j \sqrt{f_\theta^{(2)}(x_i)} - \phi_\xi^i \right\|_2^{\beta-1} \nabla_\theta f_\theta^{(1)}(x_i)$$

while the formula for the variance is:

$$
\frac{\partial}{\partial \sigma^2} \hat{S}(P_\theta, P_\xi) = \frac{\partial}{\partial \sigma^2} \frac{1}{N} \sum_{i=1}^{N} \left[ \frac{2}{r} \sum_{j=1}^{r} \left\| \mu_i + \epsilon_j \sqrt{\sigma_i^2} - \phi_\xi^i \right\|_2^\beta \right.
$$

$$
\left. - \frac{1}{r(r-1)} \sum_{j,k=1, j \neq k} \left\| \mu_i + \epsilon_j \sqrt{\sigma_i^2} - \mu_i + \epsilon_k \sqrt{\sigma_i^2} \right\|_2^\beta \right]
$$

$$
= \left[ \frac{\beta}{N} \sum_{i=1}^{N} \left[ \frac{2}{r} \sum_{j=1}^{r} \left\| f_\theta^{(1)}(x_i) + \epsilon_j \sqrt{f_\theta^{(2)}(x_i)} - \phi_\xi^i \right\|_2^{\beta-1} \right. \right.
$$

$$
\left. \left. - \frac{1}{r(r-1)} \sum_{j,k=1, j \neq k} \left\| \sqrt{f_\theta^{(2)}(x_i)} \, (\epsilon_j - \epsilon_k) \right\|_2^{\beta-1} \right] \right] \nabla_\theta f_\theta^{(2)}(x_i)
$$

### 10.5.4 EXCHANGEABILITY OF GRADIENT AND EXPECTATION

This proof follows a similar argumentation as the proof performed in Pacchiardi & Dutta (2022). We want to solve

$$
\hat{\theta} := \arg\min_\theta J(\theta), \quad J(\theta) = \frac{1}{N} \sum_{i=1}^{N} S(P_\theta, z_\xi^i) \tag{10}
$$

Finding a minimum in our network architecture is done via stochastic gradient descent (SGD) or an algorithm exploiting the same properties as SGD. Recall that the scoring rule is defined as an expectation over samples from $P_\theta$. Additionally, we can describe the scoring rule as some function $h$ of independent inputs $z_j^i \perp z_k^i$. Hence we can write:

$$
S(P_\theta, z_\xi^i) = \mathbb{E}_{z_j^i, z_k^i \sim P_\theta} \left[ h\left( z_j^i, z_k^i, z_\xi^i \right) \right] \tag{11}
$$

Let $\mu_i, \sigma_i^2$ be the estimated mean and variance, respectively, of the online network $f_\theta$ for the $i$-th input sample $x_i$. With $P_\theta$ being the distribution induced by the online network and $g_\theta$ being its transformation, we use the reparametrization trick and get

$$
J(\theta) = \frac{1}{N} \sum_{i=1}^{N} \mathbb{E}_{\epsilon_j^i, \epsilon_k^i \sim \mathcal{N}(0,1)} \left[ h(g_\theta(\epsilon_j^i, \mu_i, \sigma_i^2), g_\theta(\epsilon_k^i, \mu_i, \sigma_i^2), z_\xi^i) \right] \tag{12}
$$

We now can write the derivative as follows:

$$
\nabla_\theta J(\theta) = \nabla_\theta \frac{1}{N} \sum_{i=1}^{N} \mathbb{E}_{\epsilon_j^i, \epsilon_k^i \sim \mathcal{N}(0,1)} \left[ h(g_\theta(\epsilon_j^i, \mu_i, \sigma_i^2), g_\theta(\epsilon_k^i, \mu_i, \sigma_i^2), z_\xi^i) \right]
$$

$$
= \frac{1}{n} \sum_{i=1}^{n} \mathbb{E}_{\epsilon_j^i, \epsilon_k^i \sim \mathcal{N}(0,1)} \left[ \nabla_\theta h(g_\theta(\epsilon_j^i, \mu_i, \sigma_i^2), g_\theta(\epsilon_k^i, \mu_i, \sigma_i^2), z_\xi^i) \right]
$$

The exchange between expectation and gradient is not trivial because of the non-differentiability of functions (e.g., ReLU) within the network function $g_\theta$. We can still perform this step by using Theorem 5 from Pacchiardi & Dutta (2022), considering mild conditions on the neural network architecture of $g_\theta$.

### 10.5.5 UNBIASED ESTIMATE OF THE GRADIENT

Let $\mathcal{B}$ be a random subset (batch) of the data set. Using 10.5.4, we can obtain unbiased estimates of $\nabla_\theta J(\theta)$ using samples $\epsilon_i^j \sim \mathcal{N}(0,1), j = 1, ..., r$ for each $i \in \{1, ..., N\}$:

$$
\widehat{\nabla_\theta J(\theta)} = \frac{1}{|\mathcal{B}|} \sum_{i \in \mathcal{B}} \frac{1}{r(r-1)} \sum_{j,k=1, j \neq k}^{r} \nabla_\theta h(g_\theta(\epsilon_i^k, \mu_i, \sigma_i^2), g_\theta(\epsilon_i^j, \mu_i, \sigma_i^2), z_\xi^i) \tag{13}
$$

### 10.5.6 DETAILED IMPLEMENTATION

In addition to the detail in Section 5, it's important to mention that the target network takes two global augmented samples while the online network takes two global augmented samples and 16 local augmentations for multi-cropping samples. Then, the $r$ in Eq. 2 refers to the number of augmentation samples multiplied by the number of samples. Furthermore, in Eq. 2 $z_\xi$ denotes one of the global augmented samples.

## 11 ADDITIONAL EXPERIMENTAL RESULTS AND ABLATION ANALYSIS

In this section, we provide additional details on experiments and ablation analysis.

### 11.1 TRANSFER LEARNING EVALUATION

We further assess the generalization capacity of the learned representation on learning a new dataset. We followed the same transfer learning protocol explained in (Caron et al., 2021). To this end, we evaluate the performance of our model pretrained on ImageNet to CIFAR10/100, an imbalanced naturalist dataset (iNat-18), and the Flower dataset. According to the results shown in Table 4, we observe that our method provides a robust solution when transferring to the new dataset.

Table 4: **Transfer to new dataset evaluation**: Transfer learning by finetuning pretrained models on different datasets. We report top-1 accuracy. Self-supervised pretraining with ProSMin transfers better than supervised pretraining.

| Method | CIFAR-10 | CIFAR-100 | iNat-18 | Flowers |
|---|---|---|---|---|
| DINO Caron et al. (2021) | 99.0 | 90.5 | 72.0 | 98.5 |
| i-BOT Zhou et al. (2021) | 99.1 | 90.7 | 73.7 | 98.6 |
| ProSMin (ours) | 99.3 | 91.2 | 74.4 | 99.1 |

### 11.2 OUT-OF-DISTRIBUTION DETECTION AND CORRUPTED DATASET EVALUATION

Table 5 compares the performance of our method for the task of OOD detection and corrupted dataset on ImageNet-O and ImageNet-C respectively.

Table 5: **Out of distribution detection and corrupted dataset**: AUROC for out of distribution detection for **Imagent-O** dataset, where higher AUROC is better, and mCE for **Imagent-C** dataset where lower mCE is better.

| Method | AUROC (%) (↑) | mCE (↑) |
|---|---|---|
| DINO (300 epochs) Caron et al. (2021) | 74.0 | 68.1 |
| MOCO-V3 (300 epochs) Chen et al. (2021) | 71.0 | 73.5 |
| ProSMin (300 epochs) | 76.0 | 66.1 |
| DINO (800 epochs) Caron et al. (2021) | 76.0 | 66.5 |
| ProSMin (800 epochs) Caron et al. (2021) | 77.1 | 65.8 |
| i-BOT (800 epochs) Caron et al. (2021) | 76.6 | 66.2 |

### 11.3 DIMENSION OF REPRESENTATION VECTOR

The dimensionality of latent representations influences the equilibrium between information preservation, transferability, computational efficiency, and defense against overfitting. In Table 6, we evaluate the effect of varying the output dimensionality. Based on the results shown in Table 6 the large output dimensionality improves the performance.

Table 6: **Analysis of dimension of representation vector in term of Top-1 K-NN.**

| Method | 1024 | 4096 | 16384 | 65536 | 262144 |
|---|---|---|---|---|---|
| DINO Caron et al. (2021) | 67.8 | 69.3 | 69.2 | 69.7 | 69.1 |
| ProSMin (ours) | 69.0 | 69.4 | 69.5 | 69.9 | - |

## 11.4 IMPACT OF PROSMIN ON MITIGATING REPRESENTATION COLLAPSE

As we explained in Chapter 3, ProSMin formulates the collapsing representation problem through a probabilistic lens, aiming to provide a comprehensive and nuanced solution that not only addresses the limitations of deterministic approaches but also harnesses the power of uncertainty quantification and broader representation distributions. Specifically, we propose a novel probabilistic self-supervised learning method that minimizes a scoring rule during pretext task learning. We motivate ProSMin by formulating knowledge distillation (KD) in a probabilistic manner. Through extensive empirical analysis, we validate the effectiveness of our approach compared to other self-supervised approaches proposed to prevent collapsing representation (MoCoV3 (Chen et al., 2021), DINO (Caron et al., 2021), and BYOL (Grill et al., 2020).) In particular, BYOL prevents collapsing representation using batch normalization and a prediction layer. For DINO, Sharpening and centering play the same role. We showed the superiority of our method compared to other approaches for collapsing representation in Tables 1, 2, 5, and 4 as well as Figure 2 for the task of in-domain generalization, low-shot learning, semi-supervised learning, transfer learning, OOD detection, and corrupted dataset evaluation respectively. Importantly, in Table 3, we showed that our method can achieve a good performance without BN, prediction layer, centering, and sharpening which are the components of BYOL and DINO to prevent collapsing representation.

## 11.5 ANALYSIS OF COMPUTATIONAL COST

We further assess the effectiveness of our proposed approach and compare it with DINO and iBOT in Table 7. The presented values are obtained from the data reported in the DINO and iBOT papers. However, these papers do not include information regarding the total number of parameters.

Table 7: **Evaluation of computational efficiency** We conduct a thorough analysis of the computational efficiency of our novel probabilistic approach in comparison to alternative self-supervised methods. This evaluation encompasses memory utilization and computational expenditure.

| Method (ViT-s) | parameters (M) | im/s | time/ 300-epochs (hr) | number of GPUs | memory (G) |
|---|---|---|---|---|---|
| DINO Caron et al. (2021) | 21 | 1007 | 72.6 | 16 | 15.4 |
| i-BOT Zhou et al. (2021) | 21 | 1007 | 73.8 | 16 | 19.5 |
| ProSMin (ours) | 21 | 1007 | 98.0 | 8 | 21.1 |

## 11.6 BROADER IMPACT AND LIMITATIONS

This study has the potential to inspire new algorithms and stimulate theoretical and experimental exploration. The algorithm presented here can be used for many different probabilistic downstream tasks, including (but not limited to) uncertainty quantification, density estimation, image retrieval, probabilistic unsupervised clustering, program debugging, image generation, music analysis, and ranking. In addition, we believe that our extended concept probabilistic framework opens many interesting avenues for future development in self-supervised learning, and addresses many problems of existing models, such as avoiding representation collapse.

However, there are several limitations. One limitation of our model compared to other learning methods (such as supervised learning) is that self-supervised learning may require more computational resources and training time. However, considering that our proposed method does not require manual annotation, which is usually very expensive, we would argue that this trade-off is acceptable.

.

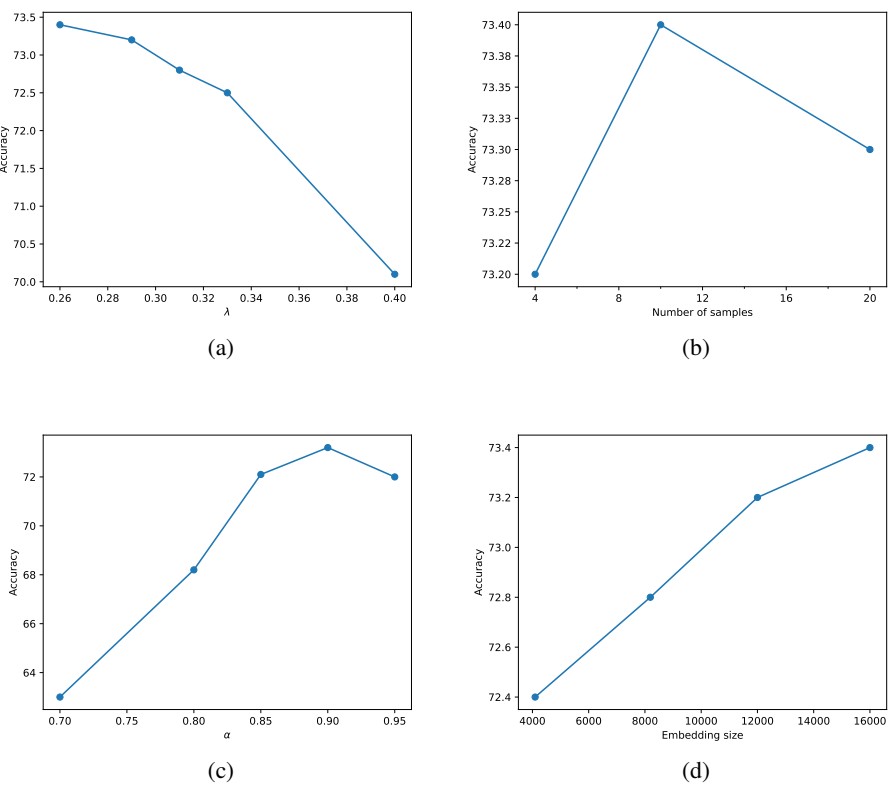

Figure 4: Study of hyperparameters of our proposed ProSMin (a) $\lambda$, (b) Number of samples, (c) Momentum coefficient ($\alpha$), and (d) Size of embedding obtained by 100 epochs on ImageNet.

