# OpenReview forum: "Probabilistic Self-supervised Representation Learning via Scoring Rules Minimization"
_ICLR.cc/2024/Conference — ICLR 2024 poster_

### Official Review · Reviewer_JrU7 · 2023-10-28

**Soundness:** 3 good
**Presentation:** 3 good
**Contribution:** 3 good
**Rating:** 6
**Confidence:** 3

**Summary:**

This paper propose ProSMin, a probabilistic self-supervised learning approach that can mitigate the problem of  collapsing representations in self-supervised learning. It follows the basic framework of training representations on augmented views of give images, like in contrastive learning. The core components are 1) an online network that predicts the representation as a distribution rather than a deterministic vector; 2) a target network that can be seen as an mean teacher.  Learning is accomplished based on proper scoring rules.

**Strengths:**

1.  Give a new probabilistic modeling for self-supervised representation learning. Compared with the deterministic modeling, the probabilistic modeling effectively mitigates collapsing representations.
2. The convergence of the proposed method is theoretically proved. The theooritical justification brings new insights on how representation quality is effectively improve
3. Solid experiments results on in-domain case and out-of-domain case show the proposed method achieves better represnetation, and the learned scores are well calibrated.

**Weaknesses:**

This paper reads smooth and everything looks good to me except for one concern: The authors are trying to sell that the proposed methods avoid the collapsing problem in self-supervised representation learning, but I didn't see explict justification or evaluation on this point.
I have no idea that representations learned with previous method, such as contrastive learning, collapsed to what extend, and no idea on how good the proposed method improved on this. Maybe the performance on in-domain and out-of-domain expereiments show the representation is better, but I cannot justify if or not this were because the collapse problem is mitigated.
I believe this work would be definitely more technically sound If authors provide more qualititive or quantatitive results that can directly reflect the level of collapse,

**Questions:**

Please see [Weakness]

---

> ### Author Response · Authors · 2023-11-19
> **Response to Reviewer JrU7**
>
> We thank the reviewer for the positive acknowledgment and valuable comments. We refer to General Response 1 for the impact of our proposed method on mitigating collapsing representation.

---

> ### Author Response · Authors · 2023-11-20
> **Follow-up**
>
> Dear Reviewer JrU7,
>
> We sincerely appreciate your valuable time and effort spent reviewing our manuscript. As the deadline for the discussion nears, we would like to ask you to participate. We just wonder whether there is any further concern and hope to have a chance to respond before the discussion phase ends.
>
> Many thanks, Authors

---

> > ### Comment · Reviewer_JrU7 · 2023-11-22
> > **Response to author rebuttal**
> >
> > Thanks for the rebuttal, I understand that various of experiments on multiple tasks, such as in-domain generalization, low-shot learning, semi-supervised learning, transfer learning, and OOD detection, show that ProSMin is superior to previous SSL representations. And the results with $\lambda=0, \sigma=0$ address some of my concerns. Therefore I remain my possitive rating.
> >
> > However, it would be better if we can see more intuitive justification on the improvement for the  mode collapse problem. Maybe some visualizations?

---

> > > ### Author Response · Authors · 2023-11-22
> > > **A new result on mitigating representation collapse!**
> > >
> > > Dear Reviewer JrU7,
> > >
> > > Following your comment, we have performed an additional experiment to show the impact of our proposed method for mitigating representation collapse. In particular, we compared the mean cosine similarity when the network is trained **with** and **without probabilistic parameters ($\lambda$=0 and $\sigma$=0)**. Cosine similarity measures the similarity between vectors representing data instances, and its mean across a set of instances can provide insight into the structure or relationships within the learned representations. It measures the cosine of the angle between two vectors and ranges from -1 (perfectly opposed) to 1 (perfectly aligned), with 0 indicating orthogonality. We observed that the mean of the cosine similarity is close to 1 when the network is trained with $\lambda$=0 and $\sigma$=0, while the mean of the cosine similarity is close to zero with probabilistic parameters. We hope these intuitive results address your concern.
> > >
> > > Thank you, Authors

---

### Official Review · Reviewer_pLHM · 2023-10-30

**Soundness:** 3 good
**Presentation:** 2 fair
**Contribution:** 3 good
**Rating:** 5
**Confidence:** 5

**Summary:**

The article focuses on the dimensional collapse problem in SSL. To address this issue, the authors propose a probabilistic approach via self-distillation to build robust representations. Detailed proofs confirm the convergence of the proposed method. The experimental results demonstrate the effectiveness of the method across various scenarios, including in-distribution, out-of-distribution, transfer learning, and more.

**Strengths:**

1. The paper proposes to learn robust feature representations through a probabilistic approach.
2. The theoretical proofs presented in the article, along with the explanations using scoring rules, demonstrate the convergence of the proposed algorithm.
3. The authors conduct experiments in multiple settings, and the experimental results validate the effectiveness of the proposed method.

**Weaknesses:**

1. There are some typos in the article, affecting the overall readability of the paper.
- In Table 3, should "PN" be replaced with "PL"? The experimental setups in the first and last rows of Table 3 are exactly the same. Is this an error?
- In the last sentence of the third paragraph in Chapter 4, should "$\mu $" and "$\sigma $" have the superscript "i"? As per my understanding, "$\mu $" and "$\sigma $" vary for each data point.
- In the eighth line of the abstract, there is a semicolon. I believe using a comma would be more appropriate.

2. The article uses a self-distillation mechanism for learning, but the description of the learning mechanism is not clear. The third paragraph of Chapter 5 contains a sentence that says, "The target network has the same backbone and projection as the online network and it learns through self-distillation". The target network is updated through the strategy of EMA, and I think the online network could be regarded as learning through self-distillation. Is there a mistake here, or is my understanding off? In addition, there are some mistakes in the descriptions of the distillation mechanism of DINO in the second paragraph of Chapter 2. The parameters of the teacher model, rather than the student model, are obtained through EMA.

3. The association between the probabilistic method and the prevention of dimensional collapse still requires further elaboration and demonstration. In the second paragraph of Chapter 1, the authors give two possible causes for the dimensional collapse. Is the author's solution inspired by either of the two causes or did it address either of the two factors? I think this needs to be further explained. I suggest two additional experiments in the following areas:
- Ablation experiments with the removal of "$\sigma $" and the second term Equation 2. I think it is necessary to illustrate the effectiveness of the probabilistic approach.
- Performance comparison with various baseline methods for the same feature dimensions.
- In Figure 3d, the dimension of the largest embedding vector is 16000, which I think is not large enough and should be further increased for experiments. As far as I know, many self-supervised methods, such as Barlow Twins, are capable of not collapsing completely in a dimension of 16000. I suggest that the authors can show that their proposed method does not collapse completely in larger dimensions. It would be better if the authors could somehow demonstrate that their method improves the effective dimensions of the learned features.

**Questions:**

1. The network structure of the proposed method is identical to BYOL. What do you think is the most important difference between these two approaches?
2. The term ${S^2}$ in Equation 4, Equation 5 and Equation 6 is only related to $\theta $. Why do you define the energy score as ${S^2}({P_\theta },{z_\xi })$?

---

> ### Author Response · Authors · 2023-11-19
> **Response to Reviewer pLHM**
>
> We thank the reviewer for the valuable comments and suggested experiments that highlighted our contributions. Please refer to General Responses 1 and 2 for W3.1 and W3.3 respectively.
>
> **Q1.** ... The most important difference between BYOL and ProSMin
>
> >* We employ a similar architecture to DINO (ViT-S) for both the encoder and the projection head, incorporating three linear layers with Gelu activation. Notably, our projection head does not include batch normalization. We then add a prediction layer similar to BYOL. Even without a batch normalization layer, we use a higher representation of 16k compared to 4096 for BYOL. However, we have shown that the performance does not suffer drastically at lower dimensions. Then, we have two layers for mean value and standard deviation. Our loss function is the proper scoring rule loss based on energy score L1 loss, and BYOL uses MSE loss. So we use a prediction layer like BYOL in terms of architecture, but we don't have batch normalization and use a different encoder and projection head, and we use another loss function based on a probabilistic approach. So, our method is different from BYOL, and we also showed that batch normalization and the prediction layer are not necessary components to prevent the collapse scenario, unlike BYOL.
>
>
>
> **Q2., W1., W2.** Suggested improvements and typos
>
> >* We thank the reviewer for pointing out the typos and we have updated our manuscript and incorporated these suggestions.
>
> **W3.** ...Is the author's solution inspired by either of the two causes or did it address either of the two factors?
> >* Thank you for asking this question, we address collapsing representation without overparametrization and heavy data augmentation instead we push the boundaries of self-supervised learning by embracing the richness of probabilistic models. By adopting a probabilistic framework, our proposed method encourages the neural network to assimilate a comprehensive distribution of representations, rather than being constrained to a singular, fixed representation.
>
> **W3.2.** Performance comparison with various baseline methods for the same feature dimensions.
>
> >* We have the experiments for different embedding sizes in Figure 3-d for 100 epochs, and we also add a new result with the same embedding size as DINO.  We summarize the knn results in the following Table:
> | Methods | embedding in the loss |knn|
> |----------:|------:| -----:|
> | iBOT | 8192 | 71.4|
> | Our method  | 8000 |71.2|
> | DINO | 65536 | 69.7|
> | Our method | 65536 |69.9|

---

> > ### Comment · Reviewer_pLHM · 2023-11-22
> > **Response to author rebuttal**
> >
> > The author has addressed some of my concerns. Some important experimental results are not provided. So, I keep my rating.

---

> > > ### Author Response · Authors · 2023-11-22
> > >
> > > Thank you for your reply. Would you please clarify which experiments we didn't perform?

---

> ### Author Response · Authors · 2023-11-20
> **Follow-up**
>
> Dear Reviewer pLHM,
>
> We sincerely appreciate your valuable time and effort spent reviewing our manuscript. As the deadline for the discussion nears, we would like to ask you to participate. We just wonder whether there is any further concern and hope to have a chance to respond before the discussion phase ends.
>
> Many thanks, Authors

---

### Official Review · Reviewer_QnX8 · 2023-10-31

**Soundness:** 3 good
**Presentation:** 4 excellent
**Contribution:** 3 good
**Rating:** 6
**Confidence:** 3

**Summary:**

The paper approaches the problem of learning self-supervised representation learning as a parametric probability density estimation problem in the representation space. The authors utilize an encoder to estimate the parameters of the distribution of the encoded sample similar to Kingma et al. They utilize self-distillation similar to BYOL and DINO where the distillation loss is a proper scoring rule for distributions.

**Strengths:**

The authors present a novel perspective toward solving the problem of learning self-supervised representations as a distribution over the representation space.
They achieve state-of-the-art performance on imagenet-1K for linear evaluation and K-nearest neighbour search.

**Weaknesses:**

See Questions

**Questions:**

1. How do you ensure $q_\theta$ is not an identity network w.r.t $\mu$ with $\sigma = 0$ since the dimensionality of $z_\theta$ and $t_\theta$
2. Since your scoring function is a variation of $L_2$ loss between $z_i$ and $z_{\xi}$, can you show that the variational encoding actually makes a difference by perturbing $t_\theta$ by a Gaussian noise of $\sigma = \epsilon$
3. The authors claim that their method is superior in terms of preventing the collapse of representation, can you provide an experiment specific to this to highlight that the representations learned by ProSMin mitigate this problem? This is important because the authors claim that strong augmentations distort the underlying distribution, hence promoting the collapse of the representations learned. If learning the representations via a parametric distribution that utilizes the resampling trick of sampling from a Gaussian distribution, is it not effectively an augmentation in the representation space?

---

> ### Author Response · Authors · 2023-11-19
> **Response to Reviewer QnX8**
>
> We thank reviewer QnX8 for the useful comments and suggested improvements. Please refer to General Response 1 for the first part of Q3.
>
> **Q1.** How do you ensure $q_{\theta}$ is not an identity network w.r.t $\mu$  with $\sigma$ = 0 since the dimensionality of  $z_{\theta}$ and $t_{\theta}$
>
> >* Thanks for pointing out this issue. To avoid this case, the $q_{\theta}$ contains two linear layers with GELU as nonlinearity and uses ReLU nonlinearity between $q_{\theta}$ and $\mu$, so $q_{\theta}$ is a nonlinear function between $t_{\theta}$ and $\mu$ in case $\sigma$ is zero.
>
> **Q2.** Since your scoring function is a variation of L2 loss between $z_i$  and $z_{\xi}$, can you show that the variational encoding actually makes a difference by perturbing $t_{\theta}$ by a Gaussian noise of ${\sigma}= \epsilon$
>
> >* The hyperparameter ${\lambda}$ is the determining factor for the effect of the perturbation. Thus, a lower value of ${\lambda}$ shows a more significant effect on the second-term portion of the loss, which is dominated by the injected Gaussian noise. Figure 3-a shows the effect of ${\lambda}$, lower ${\lambda}$ shows better performance compared to a higher value of ${\lambda}$. This means that injecting Gaussian noise improves the performance of the model. However, we adjust ${\lambda}$ to keep the loss function positive, which is essential for the proper scoring rule. Therefore, there is a limit to the lower value of ${\lambda}$, depending on the case.
>
> **Q3.2.** ...This is important because the authors claim that strong augmentations distort the underlying distribution, hence promoting the collapse of the representations learned.
>
> >* The effect of augmentation on collapsing representation has been discussed extensively for contrastive learning by Jing et al.[1].
>
> Reference:
> [1] Jing, L., Vincent, P., LeCun, Y., & Tian, Y. (2022). Understanding dimensional collapse in contrastive self-supervised learning. ICLR.

---

> > ### Comment · Reviewer_QnX8 · 2023-11-21
> > **Response to author rebuttal**
> >
> > The authors have answered questions 1, 3.2 to my satisfaction and hence I have increased my score to 6.
> > Also kindly fix the font sizes in all the figures.

---

> ### Author Response · Authors · 2023-11-20
> **Follow-up**
>
> Dear Reviewer QnX8,
>
> We sincerely appreciate your valuable time and effort spent reviewing our manuscript. As the deadline for the discussion nears, we would like to ask you to participate. We just wonder whether there is any further concern and hope to have a chance to respond before the discussion phase ends.
>
> Many thanks, Authors

---

### Author Response · Authors · 2023-11-19
**General Response**

We thank all reviewers for the time and expertise they have invested in these reviews and for their constructive feedback and positive recognition of the contribution of our work. First, we address the common issues raised by multiple reviewers, and then we respond to the points raised by individual reviewers.


## Impact of ProSMin on Mitigating Representation Collapse (General Response 1)

>*  As we explained in Chapter 3, ProSMin formulates the collapsing representation problem through a probabilistic lens, aiming to provide a comprehensive and nuanced solution that not only addresses the limitations of deterministic approaches but also harnesses the power of uncertainty quantification and broader representation distributions. Specifically, we propose a novel probabilistic self-supervised learning method that minimizes a scoring rule during pretext task learning. We motivate ProSMin by formulating knowledge distillation (KD) in a probabilistic manner. Through extensive empirical analysis, we validate the effectiveness of our approach compared to other self-supervised approaches proposed to prevent collapsing representation (MoCoV2, DINO, and BYOL.) In particular, BYOL prevents collapsing representation using batch normalization (BN) and a prediction layer. For DINO, Sharpening and centering play the same role. We showed the superiority of our method compared to other approaches for collapsing representation in Tables 1, 2, 4, and 5 as well as Figure 2 for the task of in-domain generalization, low-shot learning, semi-supervised learning, transfer learning, OOD detection, and corrupted dataset evaluation respectively.
Importantly, in Table 3, we showed that our method can achieve a good performance without BN, prediction layer, centering, and sharpening which are the components of BYOL and DINO to prevent collapsing representation.

>* In addition, as suggested by **Reviewer pLHM**, we performed a new experiment to highlight the importance of the probabilistic part of our loss function for avoiding collapsing representation ($\lambda$ and $\sigma$). We draw your attention to the results shown in the last row of Table 3: when the $\lambda$ = 0 and $\sigma$ = 0 the performance decreases by about 30\%! We have now added more explanation in this regard to Chapter 7, paragraph *“Impact of different components of the scoring rule“*.

>* We have now added a new Subsection 10.4 to our manuscript and discuss the impact of our proposed method for mitigating representation collapse.


## Dimension of Representation Vector (General Response 2)

>* The dimensionality of the latent representations affects the equilibrium between information preservation, transferability, computational efficiency, and defense against overfitting. We performed a new experiment requested by multiple reviewers and tried the embedding size of DINO (65k), and we got slightly better results than our model with 16k but with a smaller batch size due to limited memory and computational resources.  We added our results to a new Table 6 and discussed our results in new subsection 10.5 in the Appendix.

---

### Meta-Review · Area_Chair_G8Ni · 2023-12-14

**Metareview:**

While the proposed method ProSMin exhibits promising results and theoretical contributions in self-supervised representation learning, reviewers raise concerns about the lack of explicit evidence for its claimed mitigation of representation collapse (while additional experiment is provided, but need further details check if put in the main text).

The paper provides theoretical proofs and explanations using scoring rules to support the convergence of the proposed algorithm. ProSMin achieves state-of-the-art performance on Imagenet-1k in linear evaluation and K-nearest neighbor search.  Positive experimental results across multiple settings demonstrate the effectiveness of ProSMin on both in-domain and out-of-domain tasks. The learned scores by ProSMin are shown to be well-calibrated, indicating reliable confidence estimates.

**Justification For Why Not Higher Score:**

The description of ProSMin's self-distillation mechanism lacks clarity, particularly regarding the target network update process and its relation to online network learning. Typos and inconsistencies in the paper affect readability.

**Justification For Why Not Lower Score:**

By addressing these weaknesses and providing further evidence (some already provided in the replies, but need formalized them in the draft) for its claimed advantages, ProSMin is a solid contribution to the field of self-supervised representation learning, worth presenting to ICLR community.

---

### Decision · Program_Chairs · 2024-01-16

Accept (poster)